# MMDocBench: benchmarking large vision-language models for fine-grained visual document understanding

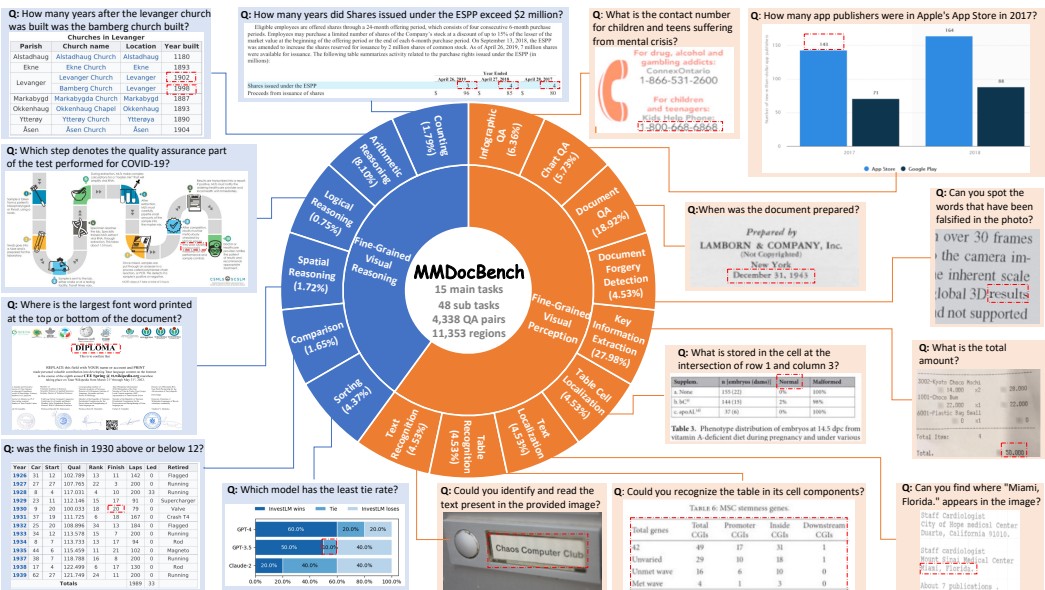

Figure 1: Overview of MMDocBench, which is designed to holistically assess the fine-grained visual understanding capability of LVLMs through various document understanding tasks. It consists of 15 main tasks and 48 sub-tasks, involving 2,400 document images, 4,338 QA pairs and 11,353 supporting regions (*i.e.,* bounding boxes). Each QA pair corresponds to one or more supporting regions, marked with red dotted-line rectangles on the images.

## Abstract

Large Vision-Language Models (LVLMs) have achieved remarkable performance in many vision-language tasks, yet their capabilities in fine-grained visual understanding remain insufficiently evaluated. Existing benchmarks either contain limited fine-grained evaluation samples that are mixed with other data, or are confined to object-level assessments in natural images. To holistically assess LVLMs' fine-grained visual understanding capabilities, we propose using document images with multi-granularity and multi-modal information to supplement natural images. In this light, we construct MMDocBench, a benchmark with various OCR-free document understanding tasks for the evaluation of fine-grained visual perception and reasoning abilities. MMDocBench defines 15 main tasks with 4,338 QA pairs and 11,353 supporting regions, covering various document images such as research papers, receipts, financial reports, Wikipedia tables, charts, and infographics. Based on MMDocBench, we conduct extensive experiments using 10 open-source and 3 proprietary advanced LVLMs, assessing their strengths and weaknesses across different tasks and document image types. The benchmark, task instructions, and evaluation code will be made publicly available.

# 1 INTRODUCTION

Large Vision-Language Models (LVLMs) have attained remarkable performance across various vision-language tasks (Yin et al., 2024). However, existing LVLMs, such as GPT-4V (OpenAI, 2023), LLaVA (Liu et al., 2023a) and MiniGPT-4 (Zhu et al., 2023), still struggle with understanding fine-grained visual details in images. For instance, (Tong et al., 2024) have demonstrated that many LVLMs perform poorly in visual grounding to image details for visual question answering. This fine-grained visual understanding capability is indispensable for LVLMs in many downstream tasks (Peng et al., 2024; Xuan et al., 2024), such as object recognition (Lin et al., 2014), image segmentation (Minaee et al., 2022) and forgery detection (Qu et al., 2023). As such, it is essential to develop LVLMs' capability in fine-grained visual understanding.

To achieve this goal, a key prerequisite is establishing benchmarks that can comprehensively evaluate the strengths and weaknesses of LVLMs in fine-grained visual understanding. However, representative multimodal benchmarks such as MMVet (Yu et al., 2023), MME (Fu et al., 2024), and MMT-Bench (Ying et al., 2024) contain relatively few data samples to examine LVLMs' understanding of fine-grained details rather than the entire image. Besides, these samples are not isolated from the overall dataset. This makes it difficult to evaluate LVLMs' ability in fine-grained visual understanding comprehensively. Moreover, some benchmarks, such as Visual7W (Zhu et al., 2016), RefCOCO (Yu et al., 2016), GVT-bench (Wang et al., 2023a), and MMBench (Liu et al., 2023b), have designed specific tasks like object grounding and object counting to evaluate fine-grained visual understanding. However, these tasks are confined to *object-level* details in natural images and do not assess finer-grained details.

To address the limitations, we consider using document images to supplement natural images for the evaluation of fine-grained visual understanding. As illustrated in Figure 1, document images encapsulate various document content types, including text, figures, tables, charts, and diagrams, all presented within a visual format. These document images like receipts and research papers are widely used across various domains such as finance, legal, education, and academia (Kim et al., 2022). Compared to natural images, document images offer certain advantages as testing data to evaluate LVLMs' fine-grained visual understanding capabilities. In particular, 1) document images contain different granularities of information to evaluate the ***fine-grained visual perception*** abilities, such as the localization and recognition of text and tables. The diverse elements (*e.g.,* text, table, and chart) tend to occupy only a small part of the entire image, yet convey critical information in various granularities; for example, a receipt image may comprise item description (sentence level), quantity (token level), and barcode (object level). Moreover, 2) document images require LVLMs to integrate multi-granularity and multi-type information to perform complex reasoning, thereby evaluating their ***fine-grained visual reasoning*** abilities. For example, in the bar chart located in the bottom left of Figure 1, LVLMs need to combine the legend, axis labels, and numerical values on the bar chart for comprehensive reasoning.

In light of this, we construct a benchmark, MMDocBench, using document images to assess LVLMs' capabilities in fine-grained visual document understanding. We design various OCR-free document understanding tasks from the perspectives of *fine-grained visual perception* and *fine-grained visual reasoning*, where understanding partial and fine-grained details in images is crucial, rather than treating the image as a whole. For *fine-grained visual perception*, MMDocBench encompasses nine tasks to evaluate the LVLMs' capabilities in fine-grained information recognition, localization, detection, and extraction, including Text Recognition, Table Recognition, Text Localization, Table Cell Localization, Key Information Extraction, Document Forgery Detection, Document Question Answering (QA), Chart QA, and Infographic QA. For *fine-grained visual reasoning*, we design six tasks to assess the reasoning ability by integrating fine-grained information: Arithmetic Reasoning, Logical Reasoning, Spatial Reasoning, Counting, Comparison, and Sorting. We present one example for each task in Figure 1 and refer to Section A.4 in Appendix for more comprehensive examples.

Based on these tasks, we select document images from 21 document understanding datasets to construct QA pairs for evaluation. The document images span a wide variety of types, including research papers, book covers, financial reports, scene-text images, receipts, Wikipedia tables, charts, infographics, and other industry documents. Additionally, in MMDocBench, we also provide annotations of supporting regions (*i.e.,* bounding boxes) within the image for each QA pair, as shown by red dotted rectangles in Figure 1. The supporting regions enable the evaluation of whether LVLMs

have correctly grounded their predictions on the associated regions in the image, leading to a more comprehensive evaluation. Furthermore, the output of supporting regions offers significant practical value, making the LVLMs' responses more informative and interpretable while allowing for rapid cross-checking between the answer and the image. Finally, MMDocBench contains $2,400$ document images, involving $4,338$ QA pairs with $11,353$ supporting regions.

With MMDocBench, we evaluate 10 open-source and 3 proprietary LVLMs that have demonstrated impressive performance in many vision-language tasks. Experimental results reveal that our MMDocBench poses significant challenges to current LVLMs, especially in region prediction, where almost all LVLMs struggle to identify the supporting regions. Although a notable gap persists in answer prediction between open-source and closed-source LVLMs, their performance in region prediction is comparable. The in-depth analyses on LVLMs' performance across different tasks and document types provide more insights: 1) Localization and detection tasks present significantly greater challenges compared to other tasks; 2) Region prediction on document images containing diverse elements (e.g., table, chart, and figure) is typically more challenging than on general documents, whereas answer prediction shows no significant variation across different document types.

In summary, our contributions are summarized as follows.

- We highlight the gap in existing benchmarks for evaluating LVLMs' fine-grained visual understanding capabilities. Besides, we propose leveraging document images with multi-granularity and multi-type information to complement natural images in assessing the fine-grained visual perception and reasoning abilities of LVLMs.

- We construct MMDocBench, a comprehensive benchmark for tracking LVLMs' progress in fine-grained visual document understanding. MMDocBench defines 15 main tasks and 48 sub-tasks over a wide range of document types, involving $2,400$ document images, $4,338$ QA pairs, and $11,353$ supporting regions for a holistic evaluation.

- We conduct extensive experiments with 13 representative LVLMs on MMDocBench. We report the major strengths and weaknesses of LVLMs in different tasks and document types and provide valuable insights, facilitating the advancements of LVLMs in future.

## 2 RELATED WORK

### 2.1 LARGE VISION-LANGUAGE MODELS

Large Vision Language Models (LVLMs) are built upon Large Language Models (LLMs) like GPTs (Radford et al., 2019; Brown et al., 2020) and LLaMA (Touvron et al., 2023). A common approach is to utilize a visual encoder to encode the image first and then apply a visual adapter to align visual and textual representations within the LLMs. LVLMs are typically trained in two stages, i.e. self-supervised pre-training over large-scale image-text pairs and supervised instruction tuning with annotated data. Notable examples include Flamingo (Alayrac et al., 2022), BLIP-2 (Li et al., 2023a), LLaVA (Liu et al., 2023a), MiniGPT-4 (Zhu et al., 2023), mPLUG-Owl (Ye et al., 2023c), CogVLM (Wang et al., 2023b), MiniCPM-V (Yao et al., 2024), Monkey (Li et al., 2024b), and InternVL (Chen et al., 2024d). For instance, BLIP-2 (Li et al., 2023a) utilizes frozen CLIP (Radford et al., 2021) as its visual encoder and proposes a Q-Former as the visual adapter. LLaVA (Li et al., 2024a) and MiniGPT-4 (Zhu et al., 2023) replace the visual adapter with simple linear layers, demonstrating impressive effectiveness.

Recently, increasing efforts have been devoted to enhancing LVLMs' performance in document image understanding (Ye et al., 2023b; Liu et al., 2024b; Ye et al., 2023a; Liu et al., 2024c; Bai et al., 2023; Zhang et al., 2023), generally by accommodating high-resolution input images or/and improving quality of training corpus. For instance, LLaVA-NeXT (Liu et al., 2024b) extends LLaVa (Liu et al., 2023a) to enhance its OCR capability by increasing image resolution and mixing more document images in visual instruction tuning data; TextMonkey (Liu et al., 2024c) and MiniCPM-V (Yao et al., 2024) divide high-resolution images into window patches with a sliding window method and reduce the length of visual tokens via token compression techniques. Compared with natural images, the understanding of document images requires LVLMs' interpretation of fine-grained image details. In this work we propose a comprehensive benchmark to evaluate such fine-grained visual understanding capability of LVLMs to facilitate future research.

Figure 2: An illustration of benchmark construction pipeline for MMDocBench.

## 2.2 EVALUATION BENCHMARKS FOR LVLMs

To date, lots of multimodal benchmarks have been constructed to holistically assess LVLMs' integrated capabilities, such as recognition, knowledge, math, reasoning and safety (Huang & Zhang, 2024). Some representative examples include LVLM-eHub (Xu et al., 2023), MME (Fu et al., 2024), MMStar (Chen et al., 2024a), SEED-Bench (Li et al., 2023b), MMMU (Yue et al., 2024a), MMMU-Pro (Yue et al., 2024b), MMVet (Yu et al., 2023) and MMT-Bench (Ying et al., 2024), etc. In addition, there are also some benchmarks designed to evaluate one specific aspect of LVLMs. For instance, POPE (Li et al., 2023b) and HallusionBench (Guan et al., 2024) focus on evaluating hallucination; MathVista (Lu et al., 2024) is centered on assessing the visual mathematical reasoning capability; OCRBench (Liu et al., 2023c) assesses the OCR capability on document images with five text-related visual tasks. These existing benchmarks include some samples that can be used to examine LVLMs' fine-grained visual understanding capability, but they are not only limited in number, but also difficult to be separated from other samples. Moreover, their evaluations rely solely on natural language output without supporting region prediction involved, which is a crucial measure for achieving fine-grained visual understanding in LVLMs (Peng et al., 2024). Current available benchmarks specially for evaluating fine-graded visual understanding of LVLMs mainly include Visual7W (Zhu et al., 2016), RefCOCO (Yu et al., 2016), GVT-bench (Wang et al., 2023a), and MMBench (Liu et al., 2023b), which are however centered on *object-level* recognition and interpretation within natural images, offering limited information granularity. Our MMDocBench is the first comprehensive benchmark aiming to evaluate LVLMs' fine-grained visual understanding capability with various OCR-free document understanding tasks.

## 2.3 VISUAL GROUNDING IN LVLMs

Visual Grounding (VG) is a technique that localizes the relevant object or region to a given natural language description in the visual input (Deng et al., 2018). It has been applied in LVLMs to facilitate generating more informative and comprehensive responses, benefiting various downstream applications like Object Recognition (Lin et al., 2014), Image Segmentation (Minaee et al., 2022) and Referential Comprehension (Yu et al., 2016). Existing grounded LVLMs can be classified into two categories: 1) region-level (i.e., bounding box) grounding based LVLMs, such as OFA (Wang et al., 2022), Kosmos-2 (Peng et al., 2024) and Pink (Xuan et al., 2024); and 2) pixel-level grounding based LVLMs, such as PixelLM (Ren et al., 2024), GlaMM (Rasheed et al., 2024), Lisa (Lai et al., 2024) and Ferret (You et al., 2024). However, the training and evaluation of these LVLMs sorely rely on *object-level* tasks involving natural images, overlooking the various granularities in document images. In this work, we build MMDocBench to foster the advancement of the fine-grained visual understanding capability in LVLMs with a variety of OCR-free document understanding tasks. In MMDocBench, it requires LVLMs to perform region-level grounding following typical settings in most document understanding tasks, such as Optical Character Recognition (OCR) (Singh et al., 2021), Key Information Extraction (Huang et al., 2019) and Table Recognition (Zheng et al., 2021).

## 3 PROPOSED MMDOCBENCH

### 3.1 PROBLEM DEFINITION

On MMDocBench, the LVLMs are expected to output the precise answer to a natural language query given a document image while highlighting the supporting regions within the image contributing to inferring the answer. Formally, given a document image $d$ possibly containing text, table, chart,

and/or figure, for a question $q$, a Large Vision-Language Model $\mathcal{M}$ is required to produce a response including the answer $a$ and the corresponding regions $R$ as supporting evidence, formulated as

$$\mathcal{M}(d, q) = (a, R). \tag{1}$$

Each region in R is a bounding box that is represented by the coordinates of its top-left and bottom-right corners in the format of $[x_1, y_1, x_2, y_2]$.

## 3.2 CONSTRUCTION PIPELINE

The pipeline for constructing our MMDocBench is illustrated in Figure 2.

**Step 1: Taxonomy Design**. Targeting at a comprehensive evaluation of LVLMs' fine-grained visual understanding capabilities, we design the taxonomy of MMDocBench following two principles.

- **Fine-grained Discrimination**: The MMDocBench must provide tasks that can adeptly evaluate LVLMs' visual comprehension capabilities with sufficient discriminability of fine-grained details in the image, rather than treating the image as a whole.

- **Diversity**: The MMDocBench must encompass a broad range of tasks in terms of required capabilities (e.g., perception, reasoning), content granularity (e.g., characters, words, tables), and document types (e.g., scientific papers, financial reports, receipts).

To solve the problem in MMDocBench, two capabilities are at the core for LVLMs, i.e. fine-grained visual perception and fine-grained visual reasoning (Liu et al., 2023b). To investigate both capabilities of LVLMs, we design a total of 15 tasks in MMDocBench. Specifically, we encompass nine tasks for fine-grained visual perception, including Text Recognition, Table Recognition, Text Localization, Table Cell Localization, Key Information Extraction, Document Forgery Detection, Document Question Answering, Chart Question Answering, and Infographic Question Answering; and for fine-grained visual reasoning, we include six tasks: Arithmetic Reasoning, Logical Reasoning, Spatial Reasoning, Comparison, Sorting and Counting. Further, we include one or multiple sub-tasks for each task to cover more diverse document image types, e.g. research papers, book covers, financial reports, scene-text images, receipts, Wikipedia tables, charts, infographics, and other industry documents, leading to a total of 48 sub-tasks in MMDocBench. See a summary in Table 1.

**Step 2: Document Image & QA Pair Preparation.** As shown in Table 1, we create BookOCR, Bbox2Text, and Text2Bbox by ourselves and use original task settings for other sub-tasks. In particular, we build BookOCR, a text recognition dataset, based on selected document images (i.e., book covers) from OCR-VQA (Mishra et al., 2019). After collecting document images, we use a pre-defined template for text recognition as the question and automatically identify all the OCR content from the image as the ground-truth answer. We build Text2Bbox and Bbox2Text with the same document images, which are selected from DocILE (Šimsa et al., 2023), RVL-CDIP (Harley et al., 2015), DocBank (Li et al., 2020), PubLayNet (Zhong et al., 2019) and PubTabNet (Zhong et al., 2020) to cover a great diversity of document types. The former task requires an LVLM to find the region in the document image given a piece of textual content, while the latter needs the model to identify corresponding text in the document image based on a specified bounding box. For the three sub-tasks, our annotators (6 undergraduate or graduate students majored in computer science) manually create one QA pair for each selected document image.

For other sub-tasks, our annotators manually analyze and select appropriate document images with annotated input-output pairs from the source dataset. After that, the input-output pairs are transformed into QA pairs following the pre-defined templates. Note that all the document images and QA pairs are selected from the test set of the source dataset except CORD (Park et al., 2019), DUDE (Van Landeghem et al., 2023) and CharXiv (Wang et al., 2024b). CORD has an insufficient number of high-quality document images in its test set, so we select some from its validation set. DUDE and CharXiv have not yet released their test sets, and thus we utilize their validation sets instead. For each sub-task, we at most select 100 document images.

It is worth mentioning that for preparing sub-tasks of fine-grained visual reasoning, we purposely select five existing datasets to ensure our MMDocBench includes a great diversity of document image types. These datasets include DUDE (Van Landeghem et al., 2023) containing general documents from various industries, WTQ (Pasupat & Liang, 2015) containing table-based documents

Table 1: Taxonomy and statistics of MMDocBench.

| Main Task | Sub Task | Document Image Type | # Images | # QA Pairs | # Regions |
|---|---|---|---|---|---|
| **Fine-Grained Visual Perception** | | | | | |
| Text Recognition | TextOCR (Singh et al., 2021) | Scene-Text Images | 100 | 100 | 100 |
| | BookOCR (Mishra et al., 2019) | Book Covers | 100 | 100 | 438 |
| Table Recognition | FinTabNet (Zheng et al., 2021) | Financial Reports | 100 | 100 | 1,864 |
| | PubTables-1M (Smock et al., 2022) | Scientific Papers | 100 | 100 | 3,520 |
| Text Localization | Text2Bbox (Šimsa et al. (2023) etc.) | Industry Documents | 100 | 100 | 100 |
| | Bbox2Text (Šimsa et al. (2023) etc.) | Industry Documents | 100 | 100 | 100 |
| Table Cell Localization | FinTabNet (Zheng et al., 2021) | Financial Reports | 100 | 100 | 100 |
| | PubTables-1M (Smock et al., 2022) | Scientific Papers | 100 | 100 | 100 |
| Key Information Extraction | SROIE (Huang et al., 2019) | Receipts | 100 | 303 | 303 |
| | WildReceipt (Sun et al., 2021) | Receipts | 100 | 512 | 512 |
| | CORD (Park et al., 2019) | Receipts | 100 | 372 | 372 |
| Doc Forgery Detection | T-SROIE (Yuxin et al., 2022) | Receipts | 100 | 100 | 286 |
| | DocTamper (Qu et al., 2023) | Cross-Domain Documents | 100 | 100 | 129 |
| Document QA | DocVQA (Mathew et al., 2021) | Industry Documents | 100 | 262 | 262 |
| | WTQ (Pasupat & Liang, 2015) | Wikipedia Tables | 100 | 351 | 351 |
| | TAT-DQA (Zhu et al., 2022) | Financial Reports | 100 | 214 | 214 |
| Chart QA | ChartQA (Masry et al., 2022) | Cross-Domain Charts | 100 | 104 | 104 |
| | CharXiv (Wang et al., 2024b) | Scientific Charts | 100 | 149 | 149 |
| Infographic QA | InfographicVQA (Mathew et al., 2022) | Infographics | 100 | 281 | 281 |
| **Fine-Grained Visual Reasoning** | | | | | |
| Arithmetic Reasoning | DUDE (Van Landeghem et al., 2023) | General Documents | 13 | 15 | 34 |
| | WTQ (Pasupat & Liang, 2015) | Wikipedia Tables | 54 | 55 | 159 |
| | TAT-DQA (Zhu et al., 2022) | Financial Table-Text Documents | 98 | 217 | 453 |
| | CharXiv (Wang et al., 2024b) | Scientific Charts | 23 | 23 | 67 |
| | InfographicVQA (Mathew et al., 2022) | Infographics | 34 | 53 | 90 |
| Logical Reasoning | DUDE (Van Landeghem et al., 2023) | General Documents | 10 | 11 | 20 |
| | WTQ (Pasupat & Liang, 2015) | Wikipedia Tables | 11 | 11 | 41 |
| | TAT-DQA (Zhu et al., 2022) | Financial Table-Text Documents | 1 | 1 | 2 |
| | CharXiv (Wang et al., 2024b) | Scientific Charts | 7 | 7 | 12 |
| | InfographicVQA (Mathew et al., 2022) | Infographics | 2 | 2 | 3 |
| Spatial Reasoning | DUDE (Van Landeghem et al., 2023) | General Documents | 38 | 41 | 43 |
| | WTQ (Pasupat & Liang, 2015) | Wikipedia Tables | 4 | 4 | 8 |
| | CharXiv (Wang et al., 2024b) | Scientific Charts | 7 | 7 | 12 |
| | InfographicVQA (Mathew et al., 2022) | Infographics | 17 | 23 | 54 |
| Comparison | DUDE (Van Landeghem et al., 2023) | Cross-Domain Documents | 3 | 3 | 6 |
| | WTQ (Pasupat & Liang, 2015) | Wikipedia Tables | 33 | 34 | 74 |
| | TAT-DQA (Zhu et al., 2022) | Financial Table-Text Documents | 10 | 10 | 30 |
| | CharXiv (Wang et al., 2024b) | Scientific Charts | 16 | 16 | 44 |
| | InfographicVQA (Mathew et al., 2022) | Infographics | 13 | 15 | 44 |
| Sorting | DUDE (Van Landeghem et al., 2023) | General Documents | 3 | 3 | 6 |
| | WTQ (Pasupat & Liang, 2015) | Wikipedia Tables | 6 | 12 | 23 |
| | TAT-DQA (Zhu et al., 2022) | Financial Table-Text Documents | 7 | 7 | 14 |
| | CharXiv (Wang et al., 2024b) | Scientific Charts | 15 | 15 | 29 |
| | InfographicVQA (Mathew et al., 2022) | Infographics | 20 | 29 | 57 |
| Counting | DUDE (Van Landeghem et al., 2023) | General Documents | 51 | 55 | 244 |
| | WTQ (Pasupat & Liang, 2015) | Wikipedia Tables | 15 | 15 | 76 |
| | TAT-DQA (Zhu et al., 2022) | Financial Table-Text Documents | 14 | 14 | 26 |
| | CharXiv (Wang et al., 2024b) | Scientific Charts | 38 | 40 | 149 |
| | InfographicVQA (Mathew et al., 2022) | Infographics | 44 | 52 | 248 |

from Wikipedia, TAT-DQA (Zhu et al., 2022) containing table-text documents from financial reports, ChartXiv (Wang et al., 2024b) containing chart-based documents from scientific papers, and InfographicVQA (Mathew et al., 2022) containing infographic-based documents.

**Step 3: Region Generation.** We generate ground truth regions for each QA pair in MMDocBench to facilitate evaluation. Similar to (Xuan et al., 2024; Chen et al., 2024b), we normalize the coordinates used to represent the bounding box to the range [0, 1000] w.r.t. the image dimensions.

For the tasks regarding fine-grained visual perception, we set the answer's location on the image as the region to be annotated, while for those regarding fine-grained visual reasoning, we annotate the locations of all supporting evidences used to infer the final answer. Specifically, we first obtain the OCR results using Google OCR service for each document image in MMDocBench. For fine-grained visual perception tasks, we automatically identify the corresponding value and its bounding box in the OCR result based on the answer. If only one value matches, we use the region of this value as the correct one; otherwise, our annotators manually review and select the appropriate region for the answer. For fine-grained visual reasoning tasks, we search for the regions for each supporting evidence if the source dataset already provides the annotation of supporting evidence,

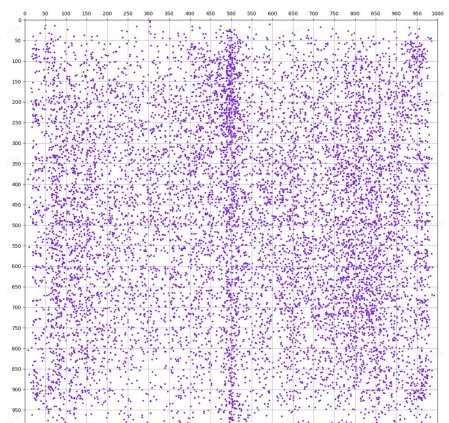 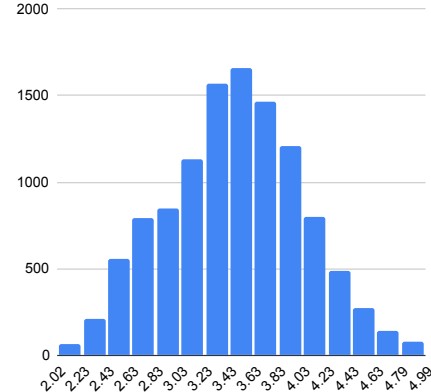

Figure 3: Position distribution of all regions.    Figure 4: Area distribution of all regions.

like TAT-DQA (Zhu et al., 2022); for the rest, our annotators manually check supporting evidence and annotate the appropriate region for each one.

**Step 4: QA & Region Verification.** To ensure high quality of MMDocBench, we further verify the collected data. First, we develop a program to automatically highlight the answer or supporting evidence with its corresponding regions on the document image. Then, different annotators review the rendered document image in three rounds to ensure that the answer and supporting regions to the question are correct.

### 3.3 STATISTICS AND ANALYSIS

With the above construction pipeline, the resultant MMDocBench contains a total of $2,400$ document images and $4,338$ QA pairs with $11,353$ annotated supporting regions. On average, each question has $10.1$ words and around $2.61$ supporting regions. Refer to Table 4 in Appendix and Table 1 for more detailed statistics of our MMDocBench.

We analyze the position distribution and area distribution of the annotated supporting regions for each QA pair in our MMDocBench. To compute the position distribution, given a region $[x_1, y_1, x_2, y_2]$, we first calculate the center point by $(\frac{x_1+x_2}{2}, \frac{y_1+y_2}{2})$. Then, we plot all the center points on an image with a dimension of $1000 \times 1000$. As shown in Figure 3, all points are scattered across the entire image, indicating no clear positional bias for supporting regions in MM-DocBench. We also analyze the area distribution of all regions to examine their granularity. We first calculate the area of each region and then apply a logarithmic transformation with a base of 10 on the computed area value. As shown in Figure 4, the granularity of regions shows a diversity and the majority of the region areas fall between $1,000$ and $10,000$, corresponding to regions sized between $10 \times 100$ and $100 \times 100$. These analyses well highlight the high quality of our MMDocBench, which is crucial for accurately assessing the capabilities of LVLMs in fine-grained visual understanding.

## 4 EXPERIMENTS

### 4.1 EXPERIMENTAL SETUP

**Evaluated LVLMs.** We conduct evaluation experiments with 10 open-source LVLMs and 3 proprietary LVLMs on the proposed MMDocBench. We select those open-source models with strong document understanding capabilities, including LLaVA-V1.6-34B (Liu et al., 2024a), Llava-OV-Chat-72B (Li et al., 2024a), TextMonkey (Liu et al., 2024c), MiniCPM-Llama3-V2.5 (Yao et al., 2024), InternVL2-8B (Chen et al., 2024c), InternVL2-Llama3-76B (Chen et al., 2024c), Qwen2-VL-7B-Instruct (Wang et al., 2024a), mPLUG-DocOwl-1.5-Omni (Hu et al., 2024), mPLUG-Owl3 (Ye et al., 2024), and Ferret (You et al., 2024). For proprietary LVLMs to be evaluated in our experiments, we use Qwen-VL-Max-0809 (Bai et al., 2023) and two versions of the latest OpenAI

Table 2: The overall performance of different LVLMs on MMDocBench. Best results are marked in bold while second-best are underlined. Metric values below 1% are marked with '-'.

| Model | Size | Fine-Grained Visual Perception | | | Fine-Grained Visual Reasoning | | | Overall | | |
|---|---|---|---|---|---|---|---|---|---|---|
| | | EM | F1 | IOU | EM | F1 | IOU | EM | F1 | IOU |
| Close-Source LVLMs | | | | | | | | | | |
| GPT-4o | - | **62.47** | 65.18 | 3.21 | **70.33** | **72.56** | 1.68 | **66.40** | 68.87 | 2.44 |
| GPT-4V | - | 52.37 | 56.95 | 2.48 | 50.97 | 52.22 | - | 51.67 | 54.58 | 1.54 |
| Qwen-VL-Max | - | 61.63 | **65.89** | 18.60 | 70.15 | 72.24 | **4.27** | 65.89 | **69.06** | **11.44** |
| Open-Source LVLMs | | | | | | | | | | |
| InternVL2-Llama3-76B | 76B | 54.63 | 58.84 | 2.30 | 61.76 | 64.46 | - | 58.20 | 61.65 | 1.54 |
| LLava-OV-Chat-72b | 72B | 52.59 | 57.49 | 2.28 | 65.26 | 67.55 | - | 58.93 | 62.52 | 1.57 |
| mPLUG-DocOwl1.5-Omni | 72B | 8.18 | 12.82 | 1.04 | 12.24 | 13.67 | - | 10.21 | 13.25 | - |
| LLaVA-V1.6-34B | 34B | 32.75 | 39.13 | 2.66 | 27.85 | 31.34 | - | 30.30 | 35.23 | 1.63 |
| Qwen2-VL-7B-Instruct | 9B | 55.15 | 59.14 | 15.16 | 43.07 | 45.88 | 2.69 | 49.11 | 52.51 | 8.93 |
| TextMonkey | 9B | 31.41 | 35.88 | **19.22** | 13.73 | 14.74 | - | 22.57 | 25.31 | 9.87 |
| MiniCPM-Llama3-V2.5 | 8B | 33.71 | 40.42 | 5.93 | 30.95 | 33.23 | 1.45 | 32.33 | 36.82 | 3.69 |
| InternVL2-8B | 8B | 45.72 | 51.42 | 2.01 | 45.16 | 48.57 | - | 45.44 | 50.00 | 1.29 |
| mPLUG-Owl3 | 7B | 15.63 | 19.97 | - | 10.84 | 12.65 | - | 13.24 | 16.31 | - |
| Ferret | 7B | 3.96 | 6.96 | 4.33 | - | - | - | 1.98 | 3.48 | 2.16 |

GPT (Achiam et al., 2023), i.e. GPT-4o-2024-08-06 and GPT-4-turbo-2024-04-09. These models show a noticeable diversity in respective parameter sizes, visual encoders, and language models.

**Instruction Design.** For each main task in MMDocBench, we manually design an instruction template to guide the LVLM to output the answer and supporting regions in JSON format, e.g., {"answer":"{answer}", "bbox":["{bbox1}","{bbox2}"]}. For fine-grained visual perception tasks, we instruct the LVLM to output the results directly, while for fine-grained visual reasoning tasks, we enable chain-of-thought (CoT) in the instructions. Since some LVLMs cannot follow the instructions well during the test, especially for the region predictions, we revise the instructions based on the setting of these LVLMs to improve their performance. At inference, we apply zero-shot prompting on all the LVLMs to generate responses for each question. After obtaining the response, we develop a program that uses strict regular expressions to extract the predicted answer and the supporting regions for evaluation.

**Evaluation Metrics.** For each question, we use *Exact Match (EM)* and $F1$-*score* to evaluate the predicted answer, and *Intersection over Union (IOU)* to assess the predicted region(s). The *EM* is determined by matching every character of the model's text prediction to the ground truth. If all characters are matched, the *EM* is 1, and otherwise 0. For $F1$-*score*, we calculate the word-level F1 based on the number of words in model prediction, ground truth, and their intersection (Rajpurkar et al., 2018). The *Intersection over Union (IOU)* is computed between the predicted region and the ground-truth region, taking into account their overlapping area and union area. The scores on the three metrics for each sub-task are computed by taking the mean of corresponding metric scores for all the questions included in this sub-task. To obtain metric scores per task or capability, and overall performance, we calculate the macro average across all corresponding lower-level metric scores.

## 4.2 MAIN RESULTS

We conduct a comprehensive comparison of different LVLMs with our proposed MMDocBench, and show the results in Table 2. We make below key findings. 1) The proposed MMDocBench poses significant challenges to current LVLMs in terms of both answer prediction and region prediction. The best model GPT-4o achieves 66.40% in EM for answer prediction, but only 2.44% in IOU for region prediction. Another close-source model, Qwen-VL-Max, which has comparable answer prediction performance to GPT-4o, is the best-performing model for region prediction, with an IOU score of 11.44% only. This highlights substantial challenges of region prediction in MMDocBench. 2) There is a large gap in answer prediction performance between open-source and close-source models, while that in their region prediction performance is minor. For answer prediction, GPT-4o significantly outperforms the best open-source model Llava-OV-Chat-72b (with 58.93% in EM) by over 12% in EM. For region prediction, open-source models like TextMonkey and Qwen2-VL-7B-Instruct, with IOU scores of 8.93% and 9.87% respectively, perform slightly worse than Qwen-VL-Max but outperform GPT-4o by around three times. This could be explained by OpenAI using insufficient samples with visual grounding requirements when training GPT-4o and GPT-4v, leading

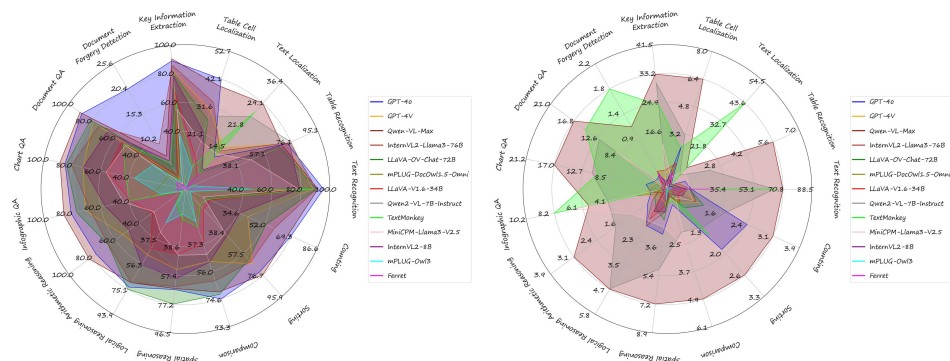

Figure 5: Model performance comparison across all tasks with F1 score.

Figure 6: Model performance comparison across all tasks with IOU metric.

to a lack of visual grounding capability. Another possible reason is that visual encoders adopted in GPT-4o and GPT-4v do not effectively support fine-grained visual understanding, which we leave for future investigation. 3) LVLMs trained on document images with text-grounding requirements, such as Qwen-VL-Max, Qwen2-VL-7B-Instruct, and TextMonkey, show improvements in region prediction, while those trained with object-level grounding over natural images, like Ferret, exhibit no such improvements. This highlights the necessity of establishing a benchmark that supports visual grounding at various and finer granularities on document images, such as our MMDocBench.

In the following, we further analyze model performance on answer prediction and region prediction regarding fine-grained visual perception and fine-grained visual reasoning, respectively.

**Answer Prediction.** We make below key findings. 1) GPT-4o consistently beats all other models on both fine-grained visual perception and fine-grained visual reasoning tasks in terms of EM, demonstrating its superior effectiveness. 2) Larger models, like GPT-4o, Qwen-VL-Max, InternVL2-Llama3-76B, and Llava-OV-Chat-72b tend to perform better on fine-grained visual reasoning tasks than fine-grained visual perception tasks, while smaller models, conversely, excel in fine-grained visual perception tasks over reasoning tasks. This might be because reasoning capabilities improve significantly as the model size increases.

**Region Prediction.** We make below key findings. 1) TextMonkey achieves the best results on fine-grained visual perception tasks with $19.22\%$ in IOU, but it fails to output supporting regions for fine-grained visual reasoning tasks. One possible reason is that TextMonkey only involves perception-related samples and instructions for training its grounding ability, resulting in its inability to ground the supporting evidence for reasoning tasks. 2) All LVLMs face significant challenges in predicting supporting regions for fine-grained visual reasoning tasks. The performance of most LVLMs on fine-grained visual reasoning tasks is notably worse than that on fine-grained visual perception tasks. Qwen-VL-Max, which is the best model for region prediction on fine-grained visual reasoning tasks, earns only $4.27\%$ in IOU, indicating the remarkable challenges of this task.

### 4.3 ANALYSIS ON DIFFERENT TASKS

We compare the performance of various LVLMs across tasks and present the results in Figure 5 and Figure 6. We make below key findings. 1) As shown in Figure 5, GPT-4o and Qwen-VL-Max are the best and second-best models across most tasks among all LVLMs, except for Logical Reasoning and Spatial Reasoning, where LLaVA-OV-Chat-72B outperforms all other models. 2) As shown in Figure 6, all LVLMs struggle with the prediction of supporting regions on almost all the tasks, except for Text Recognition, where the best model, Qwen-VL-Max, achieves around $71.2\%$ in IOU. Qwen-VL-Max delivers the best results in region prediction across most tasks, except for Forgery Detection, Infographic QA, and Text Localization, where TextMonkey ranks the first. 3) Among all the tasks, Document Forgery Detection, a fine-grained visual perception task, is the most challenging for all LVLMs, with the best result being only around $20.6\%$ in EM for answer prediction and $1.8\%$ in IOU for region prediction; As illustrated in Figure 1, this task requires the model to identify the inconsistent word(s) against other words within the image. In addition, the

Table 3: The performance comparison of LVLMs across different document types. Best results are marked in bold; second-best are underlined. Metric values below 1% are marked with a '-'.

| Model | General | | Table-Based | | Table-Text | | Chart-Based | | Infographic-Based | |
|---|---|---|---|---|---|---|---|---|---|---|
| | F1 | IOU | F1 | IOU | F1 | IOU | F1 | IOU | F1 | IOU |
| Close-Source LVLMs | | | | | | | | | | |
| GPT-4o | **71.21** | 3.25 | **71.31** | 1.39 | 77.97 | 2.90 | 71.53 | 3.06 | 74.41 | 1.38 |
| GPT-4V | 58.65 | 2.43 | 56.84 | - | 62.76 | 2.17 | 54.89 | 1.64 | 64.27 | 1.14 |
| Qwen-VL-Max | 68.45 | **19.38** | 67.35 | **5.23** | **80.89** | **9.52** | **74.18** | **11.53** | **79.41** | 4.39 |
| Open-Source LVLMs | | | | | | | | | | |
| InternVL2-Llama3-76B | 65.72 | 2.57 | 60.28 | - | 68.37 | - | 68.39 | 1.38 | 68.67 | - |
| LLaVA-OV-Chat-72B | 61.83 | 2.65 | 65.34 | - | 69.28 | 1.17 | 68.39 | 1.10 | 69.53 | - |
| mPLUG-DocOwl1.5-Omni | 17.06 | - | 13.35 | - | 11.84 | - | 12.99 | - | 19.69 | 1.25 |
| LLaVA-V1.6-34B | 42.53 | 2.57 | 33.42 | - | 36.65 | - | 40.62 | - | 33.96 | - |
| Qwen2-VL-7B-Instruct | 55.64 | 16.57 | 50.14 | 2.67 | 64.12 | 5.17 | 55.92 | 7.57 | 59.74 | 2.81 |
| TextMonkey | 36.11 | 15.44 | 20.17 | 4.01 | 24.70 | 3.34 | 23.62 | 5.51 | 27.35 | **4.43** |
| MiniCPM-Llama3-V2.5 | 40.90 | 6.67 | 34.97 | 1.59 | 41.64 | 3.26 | 42.61 | - | 39.49 | 3.83 |
| InternVL2-8B | 55.24 | 2.04 | 52.17 | - | 53.01 | - | 57.93 | - | 52.49 | - |
| mPLUG-Owl3 | 21.27 | - | 10.89 | - | 18.76 | - | 22.02 | - | 15.53 | - |
| Ferret | 8.61 | 7.17 | 3.08 | - | 4.70 | - | 5.81 | - | 5.63 | - |

answer prediction performance of all LVLMs on Text Localization and Table Cell Localization is poor, underscoring the challenges posed by both tasks. Refer to Section A.2 in Appendix for a more detailed performance comparison of all LVLMs across different tasks.

### 4.4 ANALYSIS ON DIFFERENT DOCUMENT TYPES

Based on the majority of content included, we categorize all the document images in MM-DocBench into five types: General Document, Table-Based Document, Table-Text Document, Chart-Based Document, and Infographic-Based Document, and ensure document images per sub-task belong to the same one category. Refer to Table 9 in Appendix for detailed strategies.

We present LVLMs' results across different document types in Table 3, from which we make below findings. 1) For answer prediction, GPT-4o obtains the best performance on General Document and Table-Based Document, while Qwen-VL-Max ranks the first across the other three document types. 2) For region prediction, TextMonkey outperforms all other models on Infographic-Based Document, while Qwen-VL-Max leads on the remaining four document types. 3) In comparison, region prediction on other document types is significantly more challenging than on General Documents, while there is no notable difference in answer prediction across different document types.

## 5 CONCLUSION

In this work, we introduce the MMDocBench benchmark to comprehensively evaluate LVLMs' fine-grained visual perception and reasoning capabilities via various OCR-free document understanding tasks. In MMDocBench, we carefully design 15 tasks and 48 sub-tasks that require LVLMs to perform deep, fine-grained interpretation of image details to answer each question. To enable a more comprehensive evaluation, we provide annotations of the supporting regions for each question-answer pair to assess whether LVLMs have the abilities to correctly ground their predictions on the associated regions in the image. With MMDocBench, we evaluate various open-source and proprietary LVLMs, analyzing their performance in fine-grained visual document image understanding. We observe that our MMDocBench presents significant challenges to current LVLMs in both answer and region prediction, with GPT-4o achieving the highest answer prediction score of $66.40\%$ in EM and Qwen-VL-Max achieving the best region prediction score of $11.44\%$ in IOU. Moreover, we find that open-source LVLMs demonstrate competitive performance in region prediction compared to proprietary models, despite a significant gap in answer prediction. We believe MMDocBench can enable a thorough and multi-faceted evaluation of fine-grained visual document understanding of LVLMs, thereby facilitating LVLMs' future advancement.

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

# A APPENDIX

## A.1 KEY STATISTICS

We present some key statistic about MMDocBench in Table 4.

Table 4: Key statistics of MMDocBench.

| Statistic | Number |
|---|---|
| Total Number of Tasks | 15 |
| - # Visual Perception Tasks | 9 |
| - # Visual Reasoning Tasks | 6 |
| Total Number of Sub Tasks | 48 |
| - # Visual Perception Sub-Tasks | 19 |
| - # Visual Reasoning Sub-Tasks | 29 |
| Number of Existing Datasets Involved | 21 |
| Total Number of Images | 2,400 |
| Total Number of QA Pairs | 4,338 |
| Total Number of Regions | 11,353 |
| Avg. Number of Words per Question | 16.14 |
| Avg. Number of Words per Answer | 4.08 |
| Avg. Number of Regions per Question | 2.61 |

## A.2 DETAILED PERFORMANCE OF LVLMs ACROSS DIFFERENT TASKS

Table 5: Model performance comparison across different fine-grained visual perception tasks using F1 score. TXR:Text Recognition, TBR:Table Recognition, TXL:Text Localization, TCL:Table Cell Localization, KIE:Key Information Extraction, DFD:Document Forgery Detection, DQA:Document Question Answering, CQA:Chart Question Answering, IQA:Infographic Question Answering.

| Model | Fine-Grained Visual Perception | | | | | | | | |
|---|---|---|---|---|---|---|---|---|---|
| | TXR | TBR | TXL | TCL | KIE | DFD | DQA | CQA | IQA |
| GPT-4o | 95.66 | 73.53 | 10.81 | 42.14 | 89.23 | 20.44 | 88.76 | 85.18 | 80.90 |
| GPT-4V | 88.28 | 73.33 | 14.12 | 30.04 | 82.45 | 1.99 | 79.08 | 72.31 | 70.93 |
| Qwen-VL-Max | 91.85 | 70.05 | 29.09 | 39.59 | 90.66 | 6.74 | 90.23 | 88.77 | 86.00 |
| InternVL2-Llama3-76B | 90.82 | 75.18 | 12.90 | 33.25 | 85.97 | 9.24 | 74.06 | 80.53 | 67.64 |
| LLaVA-OV-Chat-72B | 90.58 | 68.12 | 5.74 | 26.55 | 79.67 | 4.49 | 81.09 | 84.16 | 76.99 |
| mPLUG-DocOwl1.5-Omni | 33.90 | - | 0.80 | 2.19 | 7.37 | 2.25 | 15.03 | 16.73 | 24.32 |
| LLaVA-V1.6-34B | 77.68 | | 4.43 | 11.97 | 67.14 | 2.92 | 54.88 | 55.27 | 38.72 |
| Qwen2-VL-7B-Instruct | 89.04 | 76.11 | 24.57 | 28.00 | 81.63 | 0.67 | 79.23 | 78.21 | 74.76 |
| TextMonkey | 88.13 | - | 26.10 | 8.70 | 59.98 | 5.50 | 53.12 | 42.66 | 38.71 |
| MiniCPM-Llama3-V2.5 | 78.40 | 47.40 | 11.91 | 18.27 | 55.52 | 1.01 | 51.40 | 56.01 | 43.83 |
| InternVL2-8B | 79.09 | 74.93 | 6.51 | 21.70 | 78.53 | 5.76 | 67.21 | 72.26 | 56.81 |
| mPLUG-Owl3 | 45.73 | 5.31 | 0.12 | 7.67 | 38.20 | 1.59 | 28.07 | 36.01 | 17.04 |
| Ferret | 29.19 | - | 5.32 | 0.57 | 0.69 | 0.93 | 7.56 | 5.81 | 5.63 |

Table 6: Model performance comparison across different fine-grained visual perception tasks using IOU. TXR:Text Recognition, TBR:Table Recognition, TXL:Text Localization, TCL:Table Cell Localization, KIE:Key Information Extraction, DFD:Document Forgery Detection, DQA:Document Question Answering, CQA:Chart Question Answering, IQA:Infographic Question Answering.

| Model | Fine-Grained Visual Perception | | | | | | | | |
|---|---|---|---|---|---|---|---|---|---|
| | TXR | TBR | TXL | TCL | KIE | DFD | DQA | CQA | IQA |
| GPT-4o | 14.64 | 0.93 | 2.09 | 2.47 | 2.04 | 0.25 | 2.06 | 3.20 | 1.18 |
| GPT-4V | 8.66 | 0.73 | 1.55 | 1.30 | 4.42 | 0.02 | 1.66 | 2.40 | 1.57 |
| Qwen-VL-Max | 70.80 | 5.59 | 10.91 | 6.40 | 33.22 | 1.12 | 16.79 | 16.96 | 5.66 |
| InternVL2-Llama3-76B | 14.48 | 0.61 | 1.20 | 0.50 | 1.45 | 0.30 | 0.62 | 0.90 | 0.62 |
| LLaVA-OV-Chat-72B | 13.48 | 0.25 | 1.18 | 0.53 | 2.60 | 0.05 | 1.03 | 0.72 | 0.65 |
| mPLUG-DocOwl1.5-Omni | 3.49 | - | 1.01 | 0.03 | 0.30 | 0.12 | 0.93 | 0.14 | 2.27 |
| LLaVA-V1.6-34B | 11.01 | | 0.55 | 1.49 | 5.45 | 0.32 | 0.94 | 0.35 | 1.13 |
| Qwen2-VL-7B-Instruct | 65.55 | 2.23 | 7.08 | 3.12 | 30.63 | 0.00 | 12.27 | 11.28 | 4.28 |
| TextMonkey | 67.54 | - | 43.64 | 2.39 | 24.30 | 1.80 | 14.72 | 10.48 | 8.15 |
| MiniCPM-Llama3-V2.5 | 12.94 | 1.09 | 9.18 | 0.68 | 13.03 | 0.03 | 9.26 | 0.40 | 6.74 |
| InternVL2-8B | 11.86 | 0.42 | 1.03 | 1.00 | 2.34 | 0.07 | 0.58 | 0.48 | 0.34 |
| mPLUG-Owl3 | 2.03 | 0.21 | 0.02 | 0.31 | 0.39 | 0.01 | 0.15 | 0.12 | 0.19 |
| Ferret | 29.12 | - | 1.47 | 0.98 | 0.98 | 0.29 | 0.83 | 0.74 | 0.23 |

Table 7: Model performance comparison across different fine-grained visual reasoning tasks using F1.

| Model | Fine-Grained Visual Reasoning | | | | | |
|---|---|---|---|---|---|---|
| | Arithmetic Reasoning | Logical Reasoning | Spatial Reasoning | Comparison | Sorting | Counting |
| GPT-4o | 76.52 | 73.55 | 67.50 | 74.64 | 73.86 | 69.28 |
| GPT-4V | 53.36 | 49.25 | 54.08 | 49.18 | 66.57 | 40.89 |
| Qwen-VL-Max | 82.37 | 70.27 | 66.80 | 69.84 | 76.70 | 67.45 |
| InternVL2-Llama3-76B | 65.77 | 70.28 | 65.44 | 62.27 | 64.83 | 58.17 |
| LLaVA-OV-Chat-72B | 65.91 | 75.09 | 77.18 | 71.51 | 62.88 | 52.74 |
| mPLUG-DocOwl1.5-Omni | 4.92 | 11.17 | 22.14 | 18.76 | 17.30 | 7.72 |
| LLaVA-V1.6-34B | 28.37 | 44.61 | 43.46 | 37.12 | 18.10 | 16.36 |
| Qwen2-VL-7B-Instruct | 50.42 | 43.61 | 53.42 | 48.93 | 38.26 | 40.62 |
| TextMonkey | 2.11 | 10.91 | 20.38 | 25.68 | 14.37 | 14.99 |
| MiniCPM-Llama3-V2.5 | 26.73 | 32.37 | 44.40 | 44.54 | 39.57 | 11.78 |
| InternVL2-8B | 48.19 | 56.64 | 60.08 | 49.73 | 41.30 | 35.50 |
| mPLUG-Owl3 | 2.81 | 25.34 | 16.79 | 22.23 | 5.39 | 3.36 |
| Ferret | 0.00 | 0.00 | 0.00 | 0.00 | 0.00 | 0.00 |

Table 8: Model performance comparison across different fine-grained visual reasoning tasks using IOU.

| Model | Fine-Grained Visual Reasoning | | | | | |
|---|---|---|---|---|---|---|
| | Arithmetic Reasoning | Logical Reasoning | Spatial Reasoning | Comparison | Sorting | Counting |
| GPT-4o | 0.66 | 1.70 | 2.81 | 0.70 | 1.85 | 2.37 |
| GPT-4V | 0.41 | 0.44 | 0.98 | 0.44 | 0.42 | 0.91 |
| Qwen-VL-Max | 3.14 | 4.68 | 7.15 | 4.90 | 2.63 | 3.14 |
| InternVL2-Llama3-76B | 0.50 | 1.73 | 1.43 | 0.18 | 0.04 | 0.86 |
| LLaVA-OV-Chat-72B | 0.42 | 0.91 | 1.92 | 0.31 | 0.58 | 1.08 |
| mPLUG-DocOwl1.5-Omni | 0.22 | 0.02 | 0.21 | 0.13 | 0.06 | 0.17 |
| LLaVA-V1.6-34B | 0.17 | 1.39 | 1.02 | 0.28 | 0.43 | 0.32 |
| Qwen2-VL-7B-Instruct | 1.74 | 4.66 | 4.99 | 1.92 | 1.18 | 1.66 |
| TextMonkey | 0.00 | 0.00 | 1.12 | 0.52 | 1.41 | 0.00 |
| MiniCPM-Llama3-V2.5 | 1.16 | 1.49 | 2.37 | 2.64 | 0.28 | 0.77 |
| InternVL2-8B | 0.11 | 1.06 | 1.43 | 0.21 | 0.18 | 0.45 |
| mPLUG-Owl3 | 0.00 | 0.04 | 0.71 | 0.00 | 0.02 | 0.00 |
| Ferret | 0.00 | 0.00 | 0.00 | 0.00 | 0.00 | 0.00 |

## A.3 DOCUMENT CATEGORY

We follow the strategy in Table 9 to divide the document images in each sub-task into five categories, i.e., General Document, Table-Based Document, Table-Text Document, Chart-Based Document and Infographic-Based Document.

Table 9: The document image category for each sub task

| Task | Sub Task | Category |
|------|----------|----------|
| Text Recognition | TextOCR | General Document |
|  | OCR-VQA | General Document |
| Table Recognition | FinTabNet | Table-Based Document |
|  | PubTables-1M | Table-Based Document |
| Text Localization | Text2Bbox | General Document |
|  | Bbox2Text | General Document |
| Table Cell Localization | FinTabNet | Table-Based Document |
|  | PubTables-1M | Table-Based Document |
| Key Information Extraction | SROIE | General Document |
|  | WildReceipt | General Document |
|  | CORD | General Document |
| Document Forgery Detection | T-SROIE | General Document |
|  | DocTamper | General Document |
| Document Question Answering | DocVQA | General Document |
|  | WTQ | Table-Based Document |
|  | TAT-DQA | Table-Text Document |
| Chart Question Answering | ChartQA | Chart-Based Document |
|  | CharXiv | Chart-Based Document |
| Infographic Question Answering | InfographicVQA | Inforgraphic-Based Document |
| Arithmetic Reasoning | DUDE | General Document |
|  | WTQ | Table-Based Document |
|  | TAT-DQA | Table-Text Document |
|  | CharXiv | Chart-Based Document |
|  | InfographicVQA | Inforgraphic-Based Document |
| Logical Reasoning | DUDE | General Document |
|  | WTQ | Table-Based Document |
|  | TAT-DQA | Table-Text Document |
|  | CharXiv | Chart-Based Document |
|  | InfographicVQA | Inforgraphic-Based Document |
| Spatial Reasoning | DUDE | General Document |
|  | WTQ | Table-Based Document |
|  | CharXiv | Chart-Based Document |
|  | InfographicVQA | Inforgraphic-Based Documen |
| Comparison | DUDE | General Document |
|  | WTQ | Table-Based Document |
|  | TAT-DQA | Table-Text Document |
|  | CharXiv | Chart-Based Document |
|  | InfographicVQA | Inforgraphic-Based Document |
| Sorting | DUDE | General Document |
|  | WTQ | Table-Based Document |
|  | TAT-DQA | Table-Text Document |
|  | CharXiv | Chart-Based Document |
|  | InfographicVQA | Inforgraphic-Based Document |
| Counting | DUDE | General Document |
|  | WTQ | Table-Based Document |
|  | TAT-DQA | Table-Text Document |
|  | CharXiv | Chart-Based Document |
|  | InfographicVQA | Inforgraphic-Based Document |

### A.4 SAMPLES FOR EACH TASK IN MMDOCBENCH

#### A.4.1 TEXT RECOGNITION

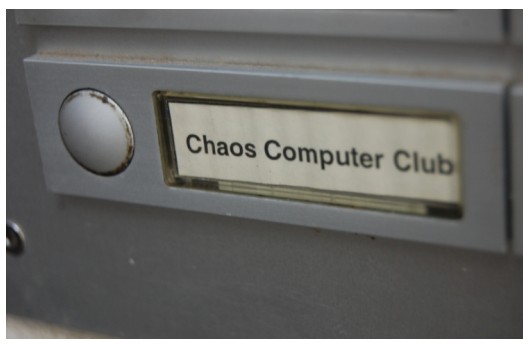

- **Question**: Could you identify and read the text present in the provided image?
- **Answer**: Chaos Computer Club
- **Bounding Box**: [339, 362, 856, 498]

#### A.4.2 TEXT LOCALIZATION

```
BIOGRAPHICAL SKETCH OF CO-INVESTIGATOR :

Name            : Uday Gadgil,M.D.
Date of Birth   : November 21,1948
Place of Birth  : Bombay, India
Citizenship     : U.S. Citizen
Work Address    : Dept of Cardiology
                  1500 E. Duarte Rd,
                  Duarte, CA 91010.

Work Phone      : (818) 359 - 8111 x 2491

Education       :
      --------------------------------------------------------
        University        Dates       Major        Degree
      --------------------------------------------------------
      University of Bombay '66 - '70   Medicine      M.D.,
      --------------------------------------------------------

MEMBERSHIP       :

Member of Los Angeles Society of Echocardiography
American Heart Association , Greater Los Angeles ,Affiliate
Fellow , American College of cardiolgy, 1978 to present

POSITIONS HELD        :

7/83 - present        : Staff Cardiologist
                        City of Hope medical Center
                        Duarte, California 91010.

7/82 - 6/83           : Staff cardiologist
                        Mount Sinai Medical Center
                        Miami, Florida.

RESEARCH PUBLICATIONS : About 7 publications .
```

- **Text2Bbox Question**: Can you find where "Miami, Florida." appears in the image and send back its position?
- **Bounding Box**: [406, 644, 576, 658]
- **Bbox2Text Question**: For the provided image and bounding box [406, 644, 576, 658], can you extract and return the text contained within the specified area?
- **Text**: Miami, Florida.

### A.4.3 TABLE RECOGNITION

TABLE 6: MSC stemness genes.

| Total genes | Total CGIs | Promoter CGIs | Inside CGIs | Downstream CGIs |
|---|---|---|---|---|
| 42 | 49 | 17 | 31 | 1 |
| Unvaried | 29 | 10 | 18 | 1 |
| Unmet wave | 16 | 6 | 10 | 0 |
| Met wave | 4 | 1 | 3 | 0 |

TABLE 7: MSC differentiation genes.

- **Question**: Could you break down the table into its cell components?
- **Answers**: [[Total genes, Total\nCGIs, Promoter\nCGIs, Inside\nCGIs, Downstream\nCGIs], [42, 49, 17, 31, 1], [Unvaried, 29, 10, 18, 1], [Unmet wave, 16, 6, 10, 0], [Met wave, 4, 1, 3, 0]]
- **Bounding Box**: [[[100,252,236,347], [320,217,385,382], [445,217,559,382], [614,217,687,382], [739,217,896,382]], [[100,394,130,482], [339,394,369,482], [489,394,513,482], [638,394,663,482], [812,394,823,482]], [[100,494,211,582], [339,494,366,582], [489,494,516,582], [638,494,663,582], [812,494,823,582]], [[100,588,252,676], [339,588,366,676], [494,588,510,676], [638,588,663,676], [809,588,823,676]], [[100,688,214,776], [345,688,361,776], [497,688,508,776], [644,688,657,776], [809,688,823,776]]]

### A.4.4 TABLE CELL LOCALIZATION

| Supplem. | n [embryos (dams)] | Normal | Malformed |
|---|---|---|---|
| a. None | 155 (22) | 0% | 100% |
| b. bC[H] | 144 (15) | 2% | 98% |
| c. apoAL[Δl] | 37 (6) | 0% | 100% |

**Table 3.** Phenotype distribution of embryos at 14.5 dpc from B vitamin A-deficient diet during pregnancy and under various re

- **Question**: What is stored in the cell at the intersection of row 1 and column 3?
- **Answer**: Normal
- **Bounding Box**: [594, 283, 684, 373]

### A.4.5 KEY INFORMATION EXTRACTION

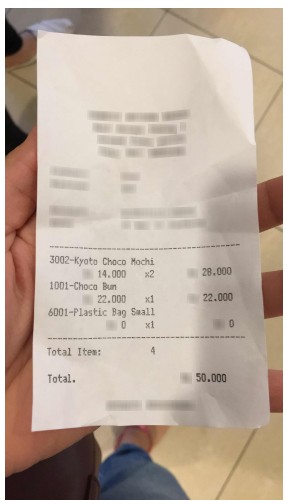

- **Question**: Please perform detection and recognition of the text that has been categorized under the "total amount" label.
- **Answer**: 50.000
- **Bounding Box**: [685, 740, 799, 762]

### A.4.6 DOCUMENT FORGERY DETECTION

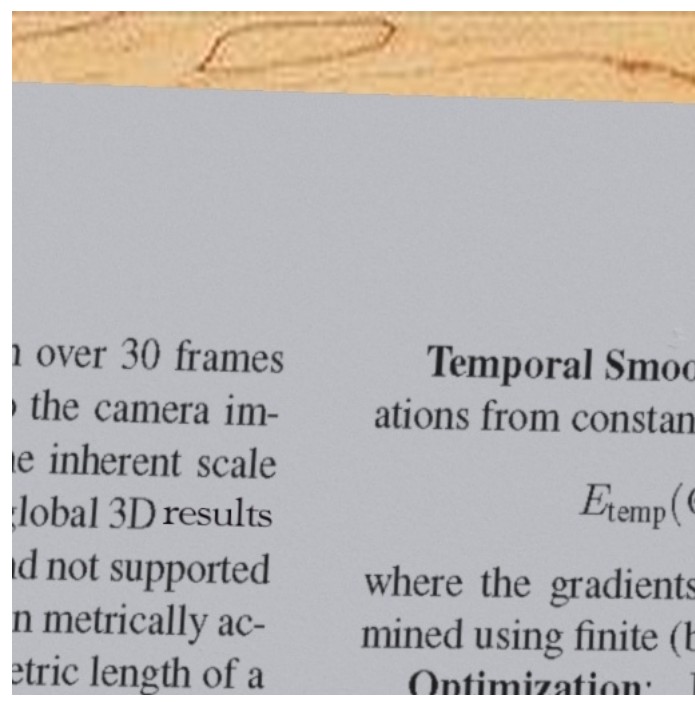

- **Question**: Can you spot the words that have been falsified in the photo?
- **Answer**: results
- **Bounding Box**: [220, 707, 380, 755]

### A.4.7 DOCUMENT QUESTION ANSWERING

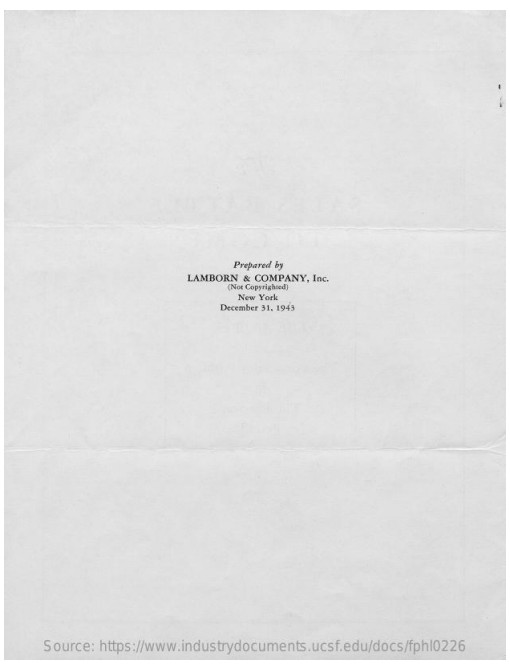

- **Question**: When was the document prepared?
- **Answer**: December 31, 1943
- **Bounding Box**: [428, 448, 579, 464]

### A.4.8 CHART QUESTION ANSWERING

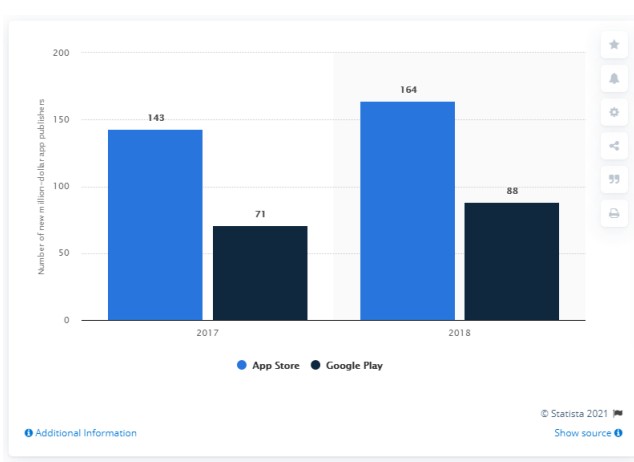

- **Question**: How many app publishers were in Apple's App Store in 2017?
- **Answer**: 143
- **Bounding Box**: [226, 222, 250, 235]

### A.4.9 INFOGRAPHIC QUESTION ANSWERING

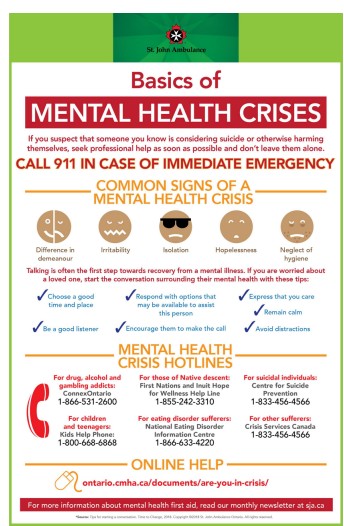

- **Question**: What is the contact number for children and teens suffering from mental crisis, 1-833-456-4566, 1-800-668-6868, or 1-866-633-4220?
- **Answer**: 1-800-668-6868
- **Bounding Box**: [147, 829, 331, 845]

### A.4.10 COUNTING

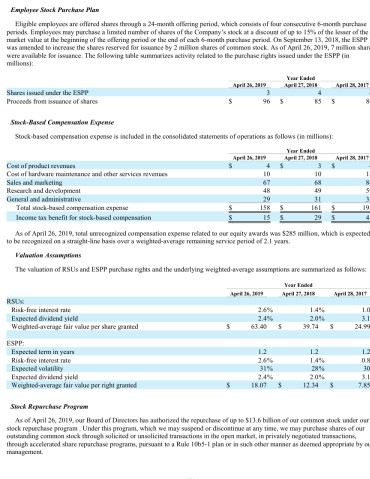

- **Question**: How many years did Shares issued under the ESPP exceed $2 million?
- **Reasoning Type**: counting
- **Answer**: 3
- **Bounding Box**: []
- **Supporting Evidence**:
    - **Text**: [4, 3, 4]
    - **Bounding Box**: [[795, 225, 807, 236], [673, 225, 685, 236], [918, 225, 929, 236]]

### A.4.11 ARITHMETIC REASONING

**Churches in Levanger**

| Parish | Church name | Location | Year built |
|---|---|---|---|
| Alstadhaug | Alstadhaug Church | Alstadhaug | 1180 |
| Ekne | Ekne Church | Ekne | 1893 |
| Levanger | Levanger Church | Levanger | 1902 |
| | Bamberg Church | Levanger | 1998 |
| Markabygd | Markabygda Church | Markabygd | 1887 |
| Okkenhaug | Okkenhaug Chapel | Okkenhaug | 1893 |
| Ytterøy | Ytterøy Church | Ytterøya | 1890 |
| Åsen | Åsen Church | Åsen | 1904 |

- **Question**: how many years after the levanger church was built was the bamberg church built?
- **Reasoning Type**: arithmetic
- **Answer**: 96
- **Bounding Box**: []
- **Supporting Evidence**:
    - **Text**: [1998, 1902]
    - **Bounding Box**: [[843, 506, 918, 551], [843, 407, 917, 452]]

### A.4.12 LOGICAL REASONING

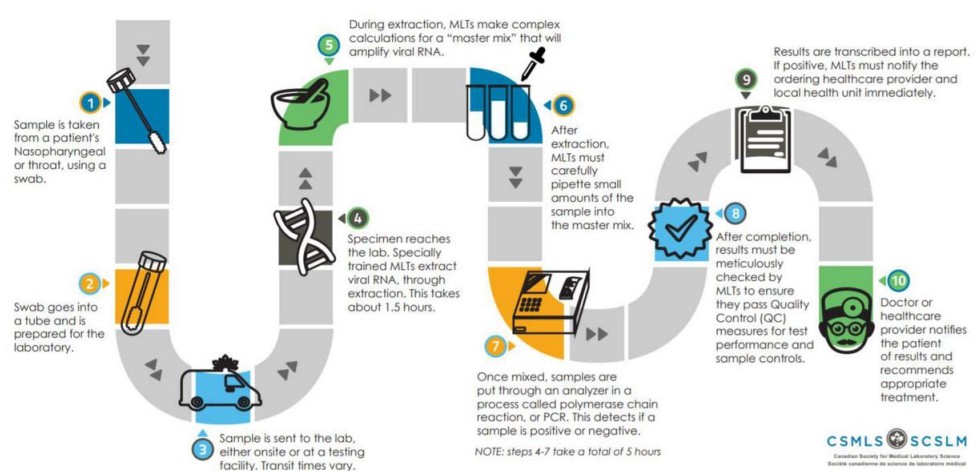

- **Question**: Which step denotes the quality assurance part of the test performed for COVID-19?
- **Reasoning Type**: logical
- **Answer**: 8
- **Bounding Box**: [726,536,744,568]
- **Supporting Evidence**:
    - **Text**: [they pass Quality Control (QC)]
    - **Bounding Box**: [[712,685,811,737]]

1404
1405
1406
1407
1408
1409
1410
1411
1412
1413
1414
1415
1416
1417
1418
1419
1420
1421
1422
1423
1424

### A.4.13 SPATIAL REASONING

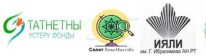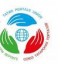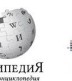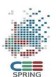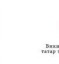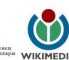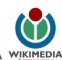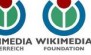

# DIPLOMA

This is to confirm that

REPLACE this field with YOUR name or account and PRINT
made personal valuable contribution into developing Tatar language content on the Internet
in the course of the eighth annual **CEE Spring @ tt.wikipedia.org** marathon
taking place on Tatar Wikipedia from March 21ˢᵗ through May 31ˢᵗ, 2022.

Co-founder and Executive
Director of Tatar Internet
Development Foundation
(Jewels of Knowledge, Literary
Marathon et al. projects)

Member of
Tatarstan Academy of Sciences,
Scientific Director of Applied Semiotics
Institute, Doctor of Technical Sciences

Corresponding member of
Tatarstan Academy of Sciences,
Director of G.Ibragimov Institute of
Language, Literature and Arts,
Doctor of Philology

Tatar Wikipedia Administrator
2018 Wikimedian of the Year
Member of Tatar & Turkic User Groups
Local Contest organizer, ttWP
representative in Tatar Portals Union

Director of «Wikimedia RU»
Non-Profit Partnership for the
Advancement of Encyclopedic
Knowledge

Director of «Belem.ru» PLC
(Tatar online content)
Member of Tatar Portals Union

TurkLang Turkic Computer Linguistics
Conference Co-Founder and Board
Member, Selet Foundation Director
General, Wiki-Collaboration initiator

Member of the Republic of Tatarstan
Presidential Commission for the
Preservation and Strengthening of
Tatar language use

Member of the Republic of Tatarstan
Presidential Commission for the
Preservation and Strengthening of Tatar
language use

Honorary Chairman of
Wikimedia Languages of Russia
volunteers community

Rail M. Gataullin

Professor Djavdet Sh. Suleymanov

Professor Kim M. Minmullin

Farhad N. Fatkullin

Vladimir V. Medeyko

8 Finalists, their qualifying contribution and awards are public at **https://w.wiki/5Heq**

- **Question**: Where is the largest font word printed at the top or bottom of the document?
- **Reasoning Type**: spatial
- **Answer**: Top
- **Bounding Box**: []
- **Supporting Evidence**:
  - **Text**: [DIPLOMA]
  - **Bounding Box**: [[382, 222, 610,268]]

### A.4.14 COMPARISON

**Churches in Levanger**

| Parish | Church name | Location | Year built |
|---|---|---|---|
| Alstadhaug | Alstadhaug Church | Alstadhaug | 1180 |
| Ekne | Ekne Church | Ekne | 1893 |
| Levanger | Levanger Church | Levanger | 1902 |
| | Bamberg Church | Levanger | 1998 |
| Markabygd | Markabygda Church | Markabygd | 1887 |
| Okkenhaug | Okkenhaug Chapel | Okkenhaug | 1893 |
| Ytterøy | Ytterøy Church | Ytterøya | 1890 |
| Åsen | Åsen Church | Åsen | 1904 |

- **Question**: was the finish in 1930 above/below 12?
- **Reasoning Type**: comparison
- **Answer**: above
- **Bounding Box**: []
- **Supporting Evidence**:
  - **Text**: [20]
  - **Bounding Box**: [[547, 358, 577, 388]]

1425
1426
1427
1428
1429
1430
1431
1432
1433
1434
1435
1436
1437
1438
1439
1440
1441
1442
1443
1444
1445
1446
1447
1448
1449
1450
1451
1452
1453
1454
1455
1456
1457

### A.4.15 SORTING

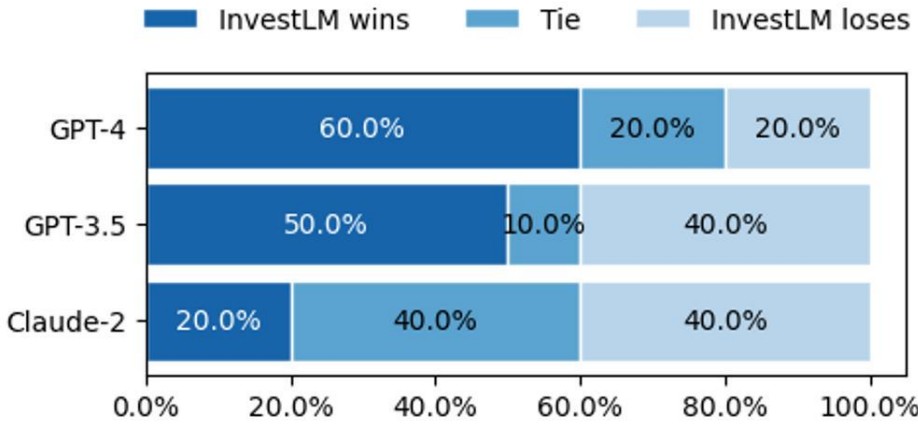

- **Question**:Which model has the least tie rate?\n * Your final answer must be grounded to some text that is explicitly written and relevant to the question in the chart.\n * If you need to answer multiple terms, separate them with commas.\n * Unless specified in the question (such as answering with a letter), you are required to answer the full names of subplots and/or labels by default.\n

- **Reasoning Type**: sorting

- **Answer**: GPT-3.5

- **Bounding Box**: [41,508,145,553]

- **Supporting Evidence**:
  - **Text**: [10.0%]
  - **Bounding Box**: [[536,493,642,575]]

