# OpenReview forum: "MMDocBench: Benchmarking Large Vision-Language Models for Fine-Grained Visual Document Understanding"
_ICLR.cc/2025/Conference — ICLR 2025 Conference Withdrawn Submission_

### Official Review · Reviewer_ne9V · 2024-10-26

**Soundness:** 3
**Presentation:** 3
**Contribution:** 2
**Rating:** 5
**Confidence:** 4

**Summary:**

This paper proposes a new benchmark for Fine-Grained Visual Document Understanding. Unlike traditional benchmarks that focus on overall image and object-level understanding, MMDocBench emphasizes evaluating the MLLM's comprehension of fine-grained image details and localized regions. MMDocBench includes a variety of sub-tasks to assess the LVLM's capability in fine-grained document understanding from multiple perspectives. Results show that both existing open-source LVLMs and proprietary LVLMs have considerable room for improvement on MMDocBench.

**Strengths:**

(1) A new fine-grained document understanding benchmark is proposed, covering 15 main tasks with 4,338 QA pairs and 11,353 supporting regions.

(2) Focusing on fine-grained image information and local regions, this benchmark increases the difficulty compared to previous benchmarks that emphasize global image information.

(3) The performance of multiple open-source and proprietary LVLMs is evaluated on this benchmark.

**Weaknesses:**

The paper lacks sufficient novelty, as there are already earlier works with similar approaches, such as MMLongBench-Doc. In the experimental section, the authors could conduct additional experiments for deeper analysis, such as incorporating human evaluation results and providing specific error case analysis.

**Questions:**

(1) The novelty of the paper is limited, as similar fine-grained document understanding benchmarks have already been introduced, such as MMLongBench-Doc. Compared to MMLongBench-Doc, what advantages does your MMDocBench offer?

(2) In the experimental section, it would be beneficial to include human performance results on MMDocBench. This could provide an additional perspective on the benchmark's difficulty.

(3) Error Analysis. Could the authors conduct a more in-depth analysis of the models' performance on MMDocBench by examining specific cases? For example, analyzing the reasons for LVLMs' reasoning errors or why different models show performance variations. Understanding the root causes behind error cases would be valuable for further improving model performance.

---

### Official Review · Reviewer_N21U · 2024-10-27

**Soundness:** 3
**Presentation:** 3
**Contribution:** 3
**Rating:** 3
**Confidence:** 5

**Summary:**

The authors provide the MMDocBench benchmark with fine-grained document understanding questions and annotations. The authors collect the images from existing benchmarks, with human-annotated QA pairs. The authors also provide the benchmark results with recent open-source and proprietary LVLMs.

**Strengths:**

1. The authors provide annotations of supporting regions, which increases the interpretability when analysis of the evaluation results. I think previous benchmarks rarely include the supporting regions.

2. The paper is easy to understand and the proposed benchmark is easy to follow.

3. The authors provide a comprehensive benchmark results of different LVLM models.

**Weaknesses:**

1. **Lack of discussion with existing document understanding benchmarks**: The motivation of this paper is fine-grained visual understanding, with a specific focus on document understanding. However, I think the existing document understanding benchmarks such as DocVQA, InfoVQA [1], DUDE, TextVQA [2], ChartQA, MMLongBench-Doc [3] have already provided the fine-grained QAs. The QAs in these benchmarks have already required the fine-grained visual understanding ability of a specific region of the whole image. Many existing LVLM models, e.g., QWen2-VL also report their results on these existing document understanding benchmarks. The authors miss the discussion/citation with [1,2,3] and do not discuss the previous document understanding benchmarks in the related work section. What's the difference between MMDocBench and these existing document understanding benchmarks? I suggest the authors provide a comparative analysis table highlighting the key differences between MMDocBench and these existing benchmarks. This could include aspects like task types, annotation methods, evaluation metrics, and any novel features of MMDocBench.
    - [1] InfographicVQA, WCAV 2022
    - [2] Towards VQA models that can read, CVPR 2019
    - [3] MMLongBench-Doc: Benchmarking Long-context Document Understanding with Visualizations, NeurIPS 2024

2. **About the LVLM baselines**: I think document understanding and OCR are hot topics in LVLM research. The related work discussed in this manuscript is mainly 2023, and many new LVLM models are not discussed. In addition, I think some of the baseline models are not chosen properly. Please refer to 2.1 - 2.3 below for details:
   - 2.1: The authors lack the discussion or benchmark results with some existing LVLMs that are specially designed for document/figure/chart/OCR understanding, such as:
     - [4] TableLlama: Towards Open Large Generalist Models for Tables, NAACL 2024
     - [5] Vary: Scaling up the Vision Vocabulary for Large Vision-Language Models, ECCV 2024
   - 2.2: In the related work section, the authors mention the LVLMs such as UReader and LLVAR that are focused on visual document understanding. Why do not report their results in Table 2?
   - 2.3: The selected proprietary models such as GPT-4o and GPT-4v are homogeneous, and the authors may consider other proprietary models such as Gemini-1.5 and Claude-3.5.

3. **The IoU score results:** According to Table 2 and lines 464-472, the authors report notably low IoU scores for all models, with GPT-4 achieving a mere 2.44% and the best-performing model reaching only 11.44%. This finding is perplexing, given the strong performance of these models on other metrics. Specifically, GPT-4, despite its low IoU score, achieves the highest EM score (66.40%) and the second-highest F1 score (68.87%). This discrepancy suggests that the IoU metric may not be the most suitable measure for evaluating region prediction in this context. I think It is unlikely that a model can consistently produce accurate answers without correctly identifying relevant regions.
    - Another possibility is that the author measured the IoU metric incorrectly. For example, the normalization of coordinates to the range [0, 1000] (lines 314-315) raises questions about the input provided to the LVLMs. But how did the models account for this coordinate normalization during region prediction? The varying image processing strategies employed by different LVLMs, such as the use of different tile sizes and resizing techniques, can impact the accuracy of coordinate transformation and region predictions. For example, 1) GPT-4o has two different image-processing strategies with different numbers of 512px tiles. 2) Some LVLM instruction following datasets (e.g., LLAVA-665k) use a different coordinate normalization method to the range [0, 1).
    - Additionally, Some LVLMs design special tokens (e.g., <|box_start|> and <|box_end|> for QWen2-VL) for referring grounding, do the authors consider that different LVLMs require different visual grounding prompt templates?
    - The authors claim that `Since some LVLMs cannot follow the instructions well during the test, especially for the region predictions, we revise the instructions based on the setting of these LVLMs to improve their performance` (Line 402), But the detailed prompts are missing and the implementation details are not clear. Do the authors think this step is fair for comparison?
    - These implementation details (input prompts, image processing strategies, special tokens, instruction following ability, and other facts that may impact the IoU score), while potentially significant, are not thoroughly discussed in the paper. The authors just claim that `All LVLMs face significant challenges in predicting supporting regions`. I think this claim is insufficient and lacks evidence.
    - I suggest the authors provide the following explanation:
      - Consider analyzing the relationship between IoU scores and answer accuracy (EM/F1-Score) to better understand this discrepancy.
      - Evaluate whether the IoU metric is the most appropriate measure for this task, or if alternative metrics might better capture region prediction performance. Does it is necessary to require the LVLMs to predict the box coordinates with the answers at the same time?
      - How the IoU metric was calculated and applied across different models.
      - Provide more implementation details, such as 1) how the coordinate normalization was communicated to the models during inference. 2) the potential impact of different image processing strategies on region prediction accuracy. 3) The details of the instruction following ability with different LVLM models and other variables that influence the results.

4. **The box prediction results are not easier for readers to assess**: The authors present the bounding box prediction results in Appendix A.4. It is hard for the reviewers to be aware of whether the box coordinates are correct or not. Can the authors provide visual examples with bounding boxes plotted on the images for a subset of representative cases? This would make it easier for readers to assess the accuracy of the predicted regions.

**Questions:**

1. Line 236: `To solve the problem in MMDocBench` -> I am wondering what the problem refers to.

2. Can the authors provide more details about the chain-of-thought (CoT) instruction in Line 402?

3. While the paper utilizes human annotators to create QA pairs (line 257), it remains unclear whether all QAs were sourced exclusively from human annotation or if some were derived from existing benchmarks like DUDE.

---

### Official Review · Reviewer_Tiou · 2024-11-02

**Soundness:** 3
**Presentation:** 3
**Contribution:** 2
**Rating:** 3
**Confidence:** 5

**Summary:**

This paper proposes a Fine-Grained Visual Document Understanding benchmark to evaluate LVLMs further. The comprehensive benchmark contains 15 main tasks across various document-level data, such as tables, figures, and so on. The authors are committed to exploring the capabilities of LVLMs for capturing visual details under more complex fine-grained visual document perception and reasoning tasks.

**Strengths:**

1. This paper is well-written and clear.
2. The grounding-like task is challenging and the authors explore further in the dense document-level scenario.

**Weaknesses:**

The differences between some document images and natural scene images require further analysis, as mentioned in the introduction.
What does ‘’fine-grained‘’ mean? If it only refers to the answer in a certain part of the image, it seems to be no different from the existing evaluation benchmark (Chart QA, DocVQA, etc).

**Questions:**

1. The density of document information is a major characteristic, but this paper does not discuss its impact on fine-grained VQA. Please investigate the impact of information density on fine-grained VQA performance in this benchmark.
2. In the cases presented in the appendix, some question areas are more obvious or structured. Are there any more dense and complex examples, such as a page of two-column paper?
3. I suggest calculating the proportion of question areas and the proportion of textual information on the image.
4. What impact does the output of the box have on hallucination problems?
5. For model design, does the author think there are any methods to improve LVLM's ability to handle fine-grained tasks？
6. Why not discuss the real fine-grained question, such as PointQA and other document-level referring tasks.

---

### Official Review · Reviewer_f3o9 · 2024-11-11

**Soundness:** 3
**Presentation:** 3
**Contribution:** 3
**Rating:** 6
**Confidence:** 2

**Summary:**

This paper introduces a benchmark named MMDocBench for OCR-free document understanding across 15 main tasks with 4,338 QA pairs and 11,353 supporting regions. Compared with existing benchmarks focusing on object-level recognition, fine-grained visual and OCR-free understanding is the main contribution. The paper evaluates 13 LVLMs on MMDocBench and provides some valuable findings.

**Strengths:**

1. Fine-grained visual and OCR-free document understanding is a good point for evaluating LVLMs.
2. The evaluated models are representative and SOTA.

**Weaknesses:**

1. There are no experiments on the instruction sensitivity, while small modifications may cause large result differences.
2. The EM metric may not work in some scenarios since LVLMs's responses are diverse.
3. The analysis of experimental results should include the model architectures, training data, and model sizes to do a more in-depth analysis. How these factors affect the results.

**Questions:**

Missing details about the CoT on instruction.

---

### Note · Authors · 2024-11-16

I have read and agree with the venue's withdrawal policy on behalf of myself and my co-authors.